# Spade-Shaped Anastomosis after Laparoscopic Proximal Gastrectomy Using Double Suture Anchoring between the Posterior Wall of the Esophagus and the Anterior Wall of the Stomach (SPADE Operation): A Case Series

**DOI:** 10.3390/cancers14020379

**Published:** 2022-01-13

**Authors:** Sin Hye Park, Harbi Khalayleh, Sung Gon Kim, Sang Soo Eom, Fahed Merei, Junsun Ryu, Young-Woo Kim

**Affiliations:** 1Center for Gastric Cancer, National Cancer Center, Ilsan-ro 323, Ilsandong-gu, Goyang-si 10408, Korea; 13606@ncc.re.kr (S.H.P.); harbikh@clalit.org.il (H.K.); kimsg0920@kyuh.ac.kr (S.G.K.); 13612@ncc.re.kr (S.S.E.); fahed.merei@ncc.re.kr (F.M.); 2Department of Otolaryngology-Head and Neck Surgery, Center for Thyroid Cancer, National Cancer Center, Ilsan-ro 323, Ilsandong-gu, Goyang-si 10408, Korea; jsryu@ncc.re.kr; 3Department of Cancer Control and Population Health, National Cancer Center Graduate School of Cancer Science and Policy, Ilsan-ro 323, Ilsandong-gu, Goyang-si 10408, Korea

**Keywords:** proximal gastrectomy, early gastric cancer, gastroesophageal reflux, laparoscopic surgery

## Abstract

**Simple Summary:**

SPADE is a novel reconstruction technique that is performed after laparoscopic proximal gastrectomy to reduce reflux. The aim of this study was to demonstrate the clinical outcomes of SPADE operations. Only one patient (2.9%) had reflux symptoms, which required anti-reflux drugs and reflux esophagitis on postoperative endoscopy. No anastomotic leakage was observed after the SPADE method. The rate of strictures at the site of anastomosis was 14.7%, and these patients were well managed with endoscopic ballooning. Therefore, the SPADE operation is a promising reconstruction method after proximal gastrectomy.

**Abstract:**

We introduced SPADE operation, a novel anastomotic method after laparoscopic proximal gastrectomy (PG). Technical modifications were performed and settled. This report aimed to demonstrate the short-term clinical outcomes after settlement. Data from 34 consecutive patients who underwent laparoscopic PG with SPADE between June 2017 and March 2020 were retrospectively reviewed. Reflux was evaluated based on the patients’ symptoms and follow-up endoscopy using Los Angeles (LA) classification and RGB Classification (Residue, Gastritis, Bile). Other complications were classified using the Clavien–Dindo method. The incidence of reflux esophagitis was 2.9% (1/34). Bile reflux was observed in six patients (17.6%), and residual food was observed in 16 patients (47.1%) in the endoscopy. Twenty-eight patients had no reflux symptoms (82.4%), while five patients (14.7%) and one patient (2.9%) had mild and moderate reflux symptoms, respectively. The rates of anastomotic stricture and ileus were 14.7% (5/34) and 11.8% (4/34), respectively. No anastomotic leakage was observed. The incidence of major complications (Clavien-Dindo grade III or higher) was 14.7%. The SPADE operation following laparoscopic PG is effective in reducing gastroesophageal reflux. Its clinical usefulness should be validated using prospective clinical trials.

## 1. Introduction

Owing to the national screening program for gastric cancer, the proportion of early gastric cancer has been recently reported to be 63.6% in Korean patients [1]. Considering the high survival rate of 97.4% for early gastric cancer, it is important to perform surgery to an optimal extent and improve the quality of life of patients with early gastric cancer [2,3]. Furthermore, similar to Western countries, the incidence of upper-third gastric cancer, especially early gastric cancer, is increasing in Asia [4,5]. In Korea, the proportion of upper-third tumors steadily increased to 20.9% in 2019 [1].

Many studies have shown no difference in survival and postoperative early complications between total gastrectomy (TG) and proximal gastrectomy (PG) [6,7]. Moreover, PG has several advantages in terms of nutritional status, reduced post-surgical weight loss, and anemia [8,9,10,11]. However, PG is not commonly performed in patients with proximal gastric cancer due to reflux [12,13]. Although the proportion of patients who underwent proximal gastrectomy has recently increased in Korea, it was still reported to be 2.6% in 2019 [1].

Various reconstruction methods, such as double tract reconstruction and the double flap method have been reported to prevent reflux [14,15]. Because these anastomosis methods are complicated, we recently devised a simple anastomosis method. The name of this method was an acronym for spade-shaped anastomosis after laparoscopic proximal gastrectomy using double suture anchoring between the posterior wall of the esophagus and the anterior wall of the stomach (SPADE), and the actual shape after anastomosis was similar to that of a spade. The keys to this operation are (1) locating the anastomosis in the abdominal cavity and never in the thoracic cavity; (2) making an artificial angle of His; (3) forming the pseudo-fornix; and (4) overlapping the esophagogastric walls around the anastomosis, which could function as a sphincter due to the distal peristaltic muscles.

The initial SPADE operation was a method of using an uncut stapler on the posterior wall of the esophagus and anterior wall of stomach, followed by hand sewing of the anterior walls of the esophagus and stomach [16]. Since June 2017, the method has been changed to hand-sew the posterior esophageal wall and the anterior gastric wall to further reduce reflux. To overcome these learning curves and to modify the method, we aimed to report the feasibility and short-term outcomes, including the occurrence of esophageal reflux of the SPADE operation.

## 2. Methods and Materials

### 2.1. Study Design and Patients

This analysis included 34 consecutive patients who recently underwent laparoscopic PG with SPADE operation using hand-sewing sutures between June 2017 and March 2020.

All patients were preoperatively diagnosed with gastric cancer using esophagogastroduodenoscopy (EGD) with biopsy and abdominopelvic computed tomography (APCT). The indications for proximal gastrectomy were clinical T1N0, T1N1, and T2N0 (c Stage Ia or Ib) in the upper third of the stomach according to the 8th edition of the TNM staging [17].

We collected data on the patients’ demographic characteristics, reflux-related clinical outcomes for 1 year after surgery, and postoperative outcome from the database of National Cancer Center, Goyang, Korea.

Postoperative complications were graded using Clavien–Dindo classification [18]. All patients were followed-up regularly according to a standardized follow-up program that consisted of physical examination, routine blood chemistry, tumor markers, chest radiography, and APCT. The reflux symptoms were classified into four categories as follows: asymptomatic; mild symptoms that did not affect quality of life and require anti-reflux medication; moderate symptoms that required medication; and severe symptoms that were inadequately improved by medication or that impeded quality of life. The patients were evaluated for reflux symptoms and prescribed anti-reflux medication in the outpatient clinic. Endoscopy was performed every 6 months or annually. The presence of anastomotic stricture was confirmed by regular follow-up endoscopy after PG. Endoscopic interventions were decided according to findings of clinical assessments at the outpatient clinic.

During follow-up endoscopy 1 year after surgery, reflux esophagitis, bile reflux, and residual food were graded according to the Los Angeles (LA) classification [19] and the RGB classification (residue, gastritis, bile) [20], respectively. This study was approved by the institutional review board of the National Cancer Center, Korea (No. NCC 2021-0161).

### 2.2. Statistical Analysis

Categorical variables were presented as frequencies and percentages, and continuous variables as means and standard deviations. If data were found to be skewed, they were presented as medians with standard deviations. All statistical analyses were performed using Systat R version 13.0 (Systat Software, Inc., San Jose, CA, USA).

### 2.3. Surgical Techniques

According to the treatment guidelines [3,21], laparoscopic PG and D1+ LN dissection was performed in all patients. The hepatic branch of the vagus nerve should be preserved in all patients.

The esophagus was divided immediately above the esophagogastric junction (EGJ), and the stomach was resected with linear staplers in consideration of the distal margin of the tumor. After confirming a negative margin on the frozen biopsy, the proximal margin of the remnant stomach and 3 cm above the cut esophagus were anchored with an interrupted suture. A linear hole with the same length as the esophageal stump was made 3 cm below the proximal margin of the anterior remnant stomach. The stapled line of the esophageal stump was resected using an energy device. Continuous layer-to-layer sutures were placed on the anterior and posterior walls of the esophagus and stomach using barbed suture material. Each barbed continuous suture was initiated on the left corner and ended on the opposite right side. After the anastomosis was completed, a spade shape was created (Figure 1).

## 3. Results

### 3.1. Patient Demographics and Clinicopathologic Findings of Patients

The patients’ background and clinicopathological outcomes are listed in Table 1. These individuals had a mean age of 60.5 years (range 33–82 years) and a body mass index of 24.6 kg/m^2^ (range 18.7–32.0 kg/m^2^). Among those patients who underwent the SPADE operation, the proportion of males was high (29/34, 85.3%). Tumor locations were common in the cardia (16/34, 47.1%) and high body (16/34, 47.1%). The median tumor size was 2.0 cm (range 1.5–2.8 cm) on preoperative endoscopy. The proportion of differentiated type was 60.6% (20/34), which was higher than that of the undifferentiated type (13/34, 39.4%). All patients who underwent the SPADE operation were diagnosed with clinical stage I, and clinical stages Ia and Ib accounted for 88.2% (30/34) and 11.8% (4/34) of cases, respectively.

### 3.2. Postoperative Endoscopic Findings and Reflux Symptom in 1 Year Follow-Up

The artificial angle of His and pseudo-fornix were well formed on postoperative endoscopy in every case (Figure 2).

Table 2 shows the postoperative endoscopic findings and reflux symptoms. Only one case of reflux esophagitis (1/34, 2.9%) was found during the 1-year follow-up endoscopy. Bile reflux and residual food were observed in 6 (6/34, 17.6%) and 16 (16/34, 47.1%) patients, respectively.

In total, 28 patients (82.4%) had no reflux symptoms, while five (5/34, 14.7%) patients and one (1/34, 2.9%) patient complained of mild and moderate symptoms, respectively.

### 3.3. Short Term Outcomes of Patients

The mean operating time was 245.4 min (range 175–340 min), and the median blood loss was 30 mL (Table 3). The median length of postoperative hospital stay was 7 days. Thirty patients (88.3%) were diagnosed with stage I disease, but four patients (11.7%) were diagnosed with Stage II in the final pathologic results.

The rate of postoperative complications was 26.5% (9/34), of which anastomotic stricture and postoperative ileus occurred in five (5/34, 14.7%) and four (4/34, 11.8%) patients, respectively. All patients with anastomotic strictures were well treated with endoscopic balloon dilatation. Postoperative ileus improved with supportive care without additional surgery. No anastomotic leakage was observed.

## 4. Discussion

In this analysis of laparoscopic PG with the SPADE operation in 34 patients with proximal EGC, there was only one case of reflux esophagitis on follow-up endoscopy. In addition, 2.9% (1/34) of patients had moderate reflux symptoms requiring anti-reflux drugs.

The SPADE methods may have resolved the physiological factors that cause reflux by maintaining the function of the fundus, preventing the creation of a sliding EGJ above the diaphragm and formation of the artificial angle of His. The angle of His between the fundus and the EGJ plays an important role in preventing natural backflow. As the anastomosis is formed in the abdominal cavity by adding two interrupted sutures (Figure 1B,C), the artificial angle of His and the pseudo-fornix were made. In addition, the overlapping of the esophageal walls in anastomosis plays an important role in reducing reflux, because it acts similarly to a sphincter by distal peristalsis. Consistent with the SPADE procedure, an artificial His angle and a pseudo-fornix were observed in all patients during follow-up endoscopy.

Double tract reconstruction has been reported to cause reflux symptoms in 1.9–4.7% and reflux esophagitis in 3.8–9.6% of cases [22,23,24,25]. In addition, the double flap method has been known to result in reflux symptoms in 6.0~7.5% and reflux esophagitis in 2.3~8.9% of patients [15,25,26,27]. The double tract method is also complicated due to the need for three anastomoses [24,28]. Furthermore, it may be less beneficial in terms of nutrition because food significantly bypasses the stomach, and surveillance of the remnant stomach is not easy due to anastomosis [14,23]. The hinged double flap method requires one esophagogastrostomy anastomosis and has the benefit of reducing reflux symptoms. However, it is difficult to perform as the double-flap method requires complicated intracorporeal suturing of the inconsequential serosal flap and longer operation time [15,26]. Therefore, the SPADE operation is effective in preventing reflux and is considered a simple procedure to compare these anastomoses.

The rate of postoperative complications related to anastomosis was 14.7%, which is similar to that of other anastomotic methods [25,29]. The review paper showed that anastomotic stricture occurred in up to 28.6% of cases, and there was no significant difference in anastomotic stricture among the reconstruction methods [29]. Therefore, the stricture rate after SPADE operation could be considered similar to other reconstruction procedures. In addition, no anastomotic leakage was observed in the present study. The rate of anastomotic leakage was reported to be 5.5% in the double-flap method and 3.8% in the double-tract method [15,22,24,26]. Therefore, the SPADE method is considered safer compared to other anastomosis methods.

Although the rate of food residue was found in 47.1% of the endoscopies, most patients did not experience symptoms related to delayed gastric emptying. The rate of residual food has been reported to be 21.8~57% after other anastomotic methods for PG. In most reports, pyloroplasty was not routinely performed to prevent dumping and duodenogastric reflux and to minimize nutritional loss [25,30,31], as in our series.

The disadvantage of the SPADE operation may be the technical aspects of anastomosis with hand sewing. Posterior wall suturing is difficult to identify. However, this could be overcome through the experience of hand sewing.

This study has several limitations. It was a small case series and reported only short-term results including reflux symptoms and anastomotic strictures after 1 year of follow-up. Additionally, the present study did not compare clinical outcomes of the SPADE method with other anastomotic methods. To prove the advantages of SPADE operation compared to other reconstruction methods, further clinical studies would be necessary. In addition, further functional and long-term studies are needed to determine nutritional variables, changes in body weight, and the degree of anemia.

## 5. Conclusions

We have presented the SPADE operation in series as a simple innovative reconstruction method that might reduce gastroesophageal reflux after laparoscopic PG. As this was a retrospective clinical study, large-scale prospective clinical studies should be conducted in the near future to validate its safety and effectiveness.

## Figures and Tables

**Figure 1 cancers-14-00379-f001:**
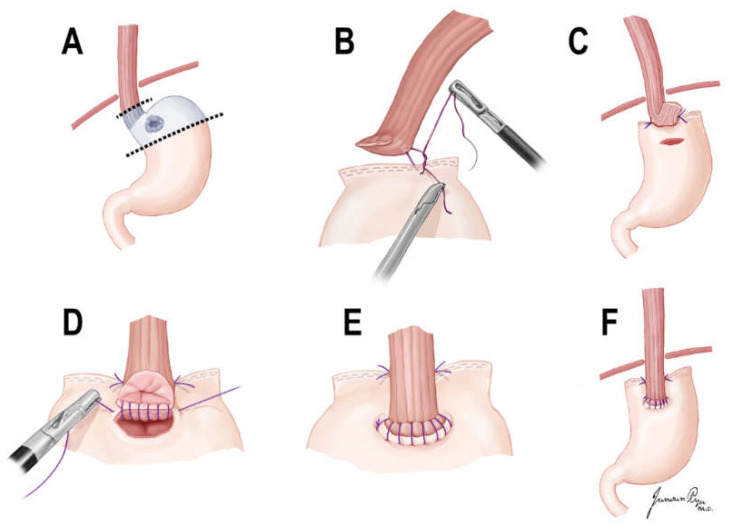
Illustration of SPADE operation. (**A**) Laparoscopic D1+ proximal gastrectomy was performed. (**B**,**C**) Both distal part of posterior wall of esophagus and proximal part of anterior wall of stomach were fixed with two interrupted sutures. (**D**) After an opening was made, one barbed continuous suture (V-Loc™) initiated at the left corner of esophagus posterior wall and stomach anterior wall, ended on the opposite right side. (**E**) After suturing of posterior wall anastomosis, anterior wall anastomosis was performed in the same maneuver. (**F**) After anastomosis was completed, a spade shape is made.

**Figure 2 cancers-14-00379-f002:**
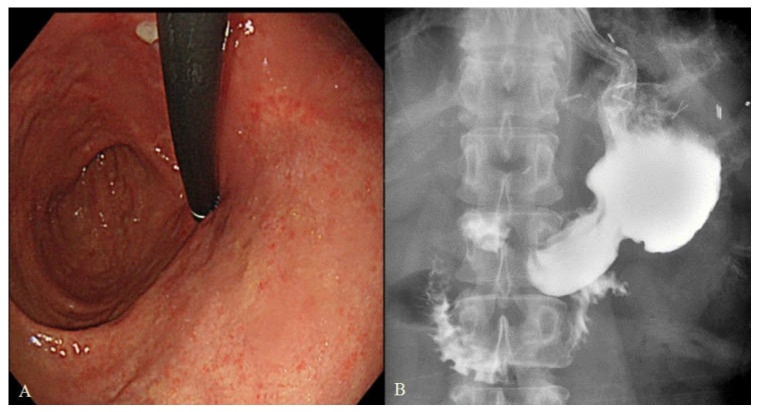
Postoperative endoscopic finding and upper gastrointestinal series after SPADE operation. (**A**) Endoscopic finding showing artificial His angle and pseudo-fornix. (**B**) Upper gastrointestinal series after SPADE method.

**Table 1 cancers-14-00379-t001:** Demographic and clinicopathological characteristics of patients.

Variable	Value [Number (%)]
Age ^†^ (year)	60.5 ± 10.7 (33–82)
BMI ^†^ (kg/m^2^)	24.6 ± 2.9 (18.7–32.0)
Sex	
Male	29 (85.3%)
Female	5 (14.7%)
Location of tumor	
Cardia	16 (47.1%)
Fundus	2 (5.9%)
High body	16 (47.1%)
Size of tumor ^‡^ (cm)	2.0 (1.5–2.8)
Histology	
WD	7 (21.2%)
MD	13 (39.4%)
PD	6 (18.2%)
SRC	7 (21.2%)
ASA score	
1	10 (29.4%)
2	20 (58.8%)
3	4 (11.8%)
c T classification	
cT1a	18 (52.9%)
cT1b	15 (44.1%)
cT2	1 (2.9%)
c N classification	
cN0	32 (94.1%)
cN1	2 (5.9%)
c Stage	
Ia	30 (88.2%)
Ib	4 (11.8%)

^†^ Values are presented as the mean ± standard deviation. ^‡^ Values are presented as median (25th and 75th percentiles). BMI: Body mass index; WD: well differentiated; MD: moderately differentiated; PD: poorly differentiated; SRC: signet ring cell.

**Table 2 cancers-14-00379-t002:** Postoperative endoscopic findings and reflux symptom in 1 year follow-up.

Variable	Value [Number (%)]
Reflux esophagitis in EGD	
No	33 (97.1%)
LA-A	0 (0%)
LA-B	1 (2.9%)
LA-C	0 (0%)
Bile reflux in EGD	
Grade 0	28 (82.4%)
Grade 1	6 (17.6%)
Residual food in EGD	
Grade 0	18 (52.9%)
Grade 1	3 (8.8%)
Grade 2	3 (8.8%)
Grade 3	10 (29.4%)
Reflux symptoms	
No symptom	28 (82.4%)
Mild symptom	5 (14.7%)
Moderate symptom	1 (2.9%)
Severe symptom	0 (0%)

EGD: esophagogastroduodenoscopy; LA: The Los Angeles Classification system.

**Table 3 cancers-14-00379-t003:** Short term outcomes of patients.

Variable	Value [Number (%)]
Operating time ^†^ (min)	245.4 ± 42.2 (175–340)
Estimated blood loss ^‡^ (ml)	30.0 (10.0–100.0)
Postoperative hospital stay ^‡^ (day)	7.0 (7.0–8.0)
Stage	
Ia	21 (61.8%)
Ib	9 (26.5%)
IIa	3 (8.8%)
IIb	1 (2.9%)
Postoperative complications	
No	25 (73.5%)
Anastomotic stricture	5 (14.7%)
Postoperative ileus	4 (11.8%)
Clavien Dindo classification	
II	4 (11.8%)
IIIa	5 (14.7%)

^†^ Values are presented as the mean ± standard deviation. ^‡^ Values are presented as median (25th and 75th percentiles).

## Data Availability

Data presented in this study are available on request from the corresponding author. The data are not publicly available because the present study was approved by the IRB that the study data should be disclosed only to the authors and not provided to third parties.

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
