# Peer review of "Spade-Shaped Anastomosis after Laparoscopic Proximal Gastrectomy Using Double Suture Anchoring between the Posterior Wall of the Esophagus and the Anterior Wall of the Stomach (SPADE Operation): A Case Series"

_cancers, 2022, doi:10.3390/cancers14020379_

Round 1

Reviewer 1 Report

Dr Park et al., presented an interesting study: ”Spade-shaped Anastomosis after Laparoscopic Proximal Gastrectomy Using Double Suture Anchoring between the Posterior Wall of the Esophagus and the Anterior Wall of the Stomach (SPADE Operation): A Case Series”

The study is well presented and well written. The surgical procedure seems very well explained and all the important variables are described. Overall, the study has a good potential. Nevertheless, some minor and major important points should be addressed before the acceptance.

Minor: Is the line 152 and 158 similar?

Minor: There is no comparison with any other technique to evaluate the present study in a robust way. It would be good to implement this aspect in the limitations.

Minor: There is not statistical comparison with the literature that can support the conclusion of the study based on previous approaches.

Minor: Overall this study does not provide enough evidence that could support the conclusions: “We noted that the SPADE operation is a feasible and simple reconstruction method 229 that can reduce gastroesophageal reflux after laparoscopic PG.” A more balanced statements in the conclusion and in the abstract is highly advised to let the reader aware of the large limitations that this study presents and to accomplish with a scientific methodology.

Reviewer 2 Report

Well written paper about a new method to prevent reflux and to prevent gastrectomy. 

The methods part and the picture ist exactly the same as in the article published J Gastric cancer 2020. https://pubmed.ncbi.nlm.nih.gov/32269846/

Although the senior author is the same - this part has to be new written ta avoid plagiarism. 

Reviewer 3 Report

The work of Park et al is novel and might represent a start-up for future multicenter studies, so that the technique will be better introduced. 

In the "Methods and Materials" section the authors mention that: All patients were preoperatively diagnosed with gastric cancer using preoperative diagnostic evaluation. The terms "diagnostic evaluation is rather unfortunate. The authors should rephrase it, and specify the order, endoscopy followed by imagistic evaluation.

Also:

The presence of anastomotic stricture was confirmed when any intervention was required to resolve the stricture.

The stricture was confirmed by performing surgery, or by endoscopy, followed by surgery? The authors should clarify this.

Author Response

  1. In the "Methods and Materials" section the authors mention that: All patients were preoperatively diagnosed with gastric cancer using preoperative diagnostic evaluation. The terms "diagnostic evaluation is rather unfortunate. The authors should rephrase it, and specify the order, endoscopy followed by imagistic evaluation.

Response)

Thank you for your kind comments. I corrected “Methods and Materials” paragraph as follows.

Original) All patients were preoperatively diagnosed with gastric cancer using preoperative diagnostic evaluation, including gastrointestinal endoscopy and abdominal computed tomography (CT).

Revised)

All patients were preoperatively diagnosed with gastric cancer using esophagogastroduodenoscopy (EGD) with biopsy and abdominopelvic computed tomography (APCT).

  1. The presence of anastomotic stricture was confirmed when any intervention was required to resolve the stricture. The stricture was confirmed by performing surgery, or by endoscopy, followed by surgery? The authors should clarify this.

Response)

Thank you for your excellent comments. I revised the sentence based on your comments

Orignial)

The presence of anastomotic stricture was confirmed when any intervention was required to resolve the stricture.

Revised)

The presence of anastomotic stricture was confirmed by regular follow-up endoscopy after proximal gastrectomy (PG).

Round 2

Reviewer 2 Report

The manuscript improved a lot. No further comments.

Reviewer 3 Report

The authors made the required changes.